# *Ex-vivo*–to–*In-vivo* Learning in Cardiology

**Alexander M. Zolotarev**[1]                     Alexander.Zolotarev@skoltech.ru
**Oleg Yu. Rogov**[1]                                         O.Rogov@skoltech.ru
**Aleksei Mikhailov**[2]                         Aleksei.Mikhailov@osumc.edu
**John D. Hummel**[2]                                     john.hummel@osumc.edu
**Vadim V. Fedorov**[2]                           Vadim.Fedorov@osumc.edu
**Dmitry V. Dylov**[1]                                         d.dylov@skoltech.ru

[1]*Skolkovo Institute of Science and Technology, 30/1 Bolshoi blvd., Moscow 121205, Russia*
[2]*The Ohio State University, Wexner Medical Center, Columbus, OH 43210, USA*

## Abstract

The clinical Atrial Fibrillation (AF) visualization method, multi-electrode mapping (MEM), delivers electrode grid *in-vivo* to the heart muscle and is known for its low resolution. A more cutting-edge imaging modality, near-infrared optical mapping (NIOM), allows seeing the AF sources in high resolution; however, it is currently *ex-vivo* only (*i.e.*, designed for explanted organs only). In this work, we present the *ex-vivo* to the *in-vivo* learning paradigm, where the former serves the purpose of improving the latter. Specifically, the NIOM improves the detection of AF sources in MEM data via an image-to-image model. We validate the idea on 7 explanted human hearts and test the models on 2 clinical cases.

**Keywords:** Atrial fibrillation, Image-to-image translation, CycleGAN, Cardiac imaging

**Introduction**   Atrial fibrillation (AF) is a serious disorder in the heart's function, characterized by an irregular rhythm of the heart's beating and a 5-fold higher risk of a stroke. Several studies indicate that the AF can be caused and maintained by localized sources in the heart tissue called AF reentrant drivers which mechanistically resemble a rotational circuit. If the AF driver's location is known, the surgeon can destroy the electrical activity in this region (the *ablation*). Unfortunately, this process is challenging due to the nature of the clinical method: the multi-electrode mapping (MEM) modality, conventionally used *in-vivo* for the AF visualization, maps a surface-only representation of the true electrical activity at a rather low resolution[1]. In contrast, another modality called near-infrared optical mapping (NIOM) relies on voltage-sensitive dyes to visualize electric patterns from the depth of the cardiac tissue. Subsurface electrical conduction, measured with high resolution[2], reveals true volumetric dynamics of the electric activity around the AF driver. Unfortunately, this method is currently available only for research experiments in *ex-vivo* setups with *explanted* organs. Recently, machine learning models were reported to correctly predict the AF driver location using MEM features validated by NIOM activation maps (Zolotarev et al., 2020).

In this study, we aspired to improve the visualisation of the AF drivers in MEM modality by image-to-image translation between MEM and NIOM maps, relying on the unpaired architecture of CycleGAN (Zhu et al., 2017). Specifically, our model learns the mapping

---

1. Standard MEM catheter (*e.g.*, FIRMap from Abbott EP, Chicago, IL) is just an 8×8 electrode grid.
2. Standard NIOM resolution is 100×100 pixels.

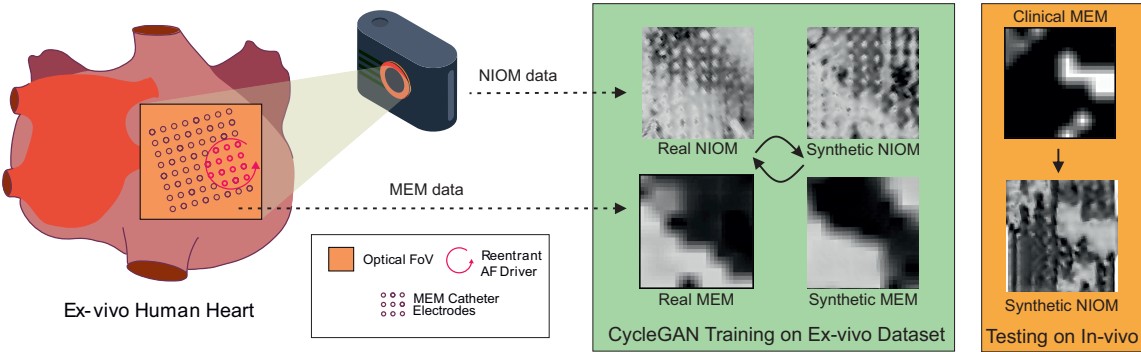

Figure 1: *Ex-vivo–*to*–in-vivo* learning paradigm for cardiac imaging. CycleGAN, pre-trained on explanted hearts, improves AF detection in clinical recordings.

between simultaneous NIOM and MEM data acquired from the human heart *ex-vivo*, and then generates synthetic NIOM images from the real MEM data. We then engage the model pre-trained on *ex-vivo* data for a limited validation on real clinical *in-vivo* recordings, providing a proof-of-principle demonstration of *ex-vivo–*to*–in-vivo* learning paradigm.

**Methods** Three CMOS cameras (spatial resolution 0.3–1.1 mm, 1000 fps, MiCAM Ultima-L, SciMedia Ltd, CA) were employed for *ex-vivo* NIOM. For clinically-relevant MEM of *ex-vivo* hearts we used 64-electrode (8×8 electrodes) catheter grids (Abbott Labs, Chicago, IL). The original methodology of experiments with NIOM and MEM can be found in (Zolotarev et al., 2020). The proposed *ex-vivo–*to*–in-vivo* learning pipeline is shown in Fig. 1.

**Datasets and Data Preprocessing** The sustained AF recordings consist of two parts: *ex-vivo* data (simultaneous MEM and NIOM movies from 7 hearts, 2 in test set) and clinically acquired *in-vivo* data (only MEM movies of 2 clinical cases). De-identified human hearts and retrospective clinical cases data were reviewed by the proper IRB. The ground-truth regions were annotated by NIOM maps for *ex-vivo* data and by the regions of successful clinical ablation for the *in-vivo* dataset. MEM recordings were analyzed by RhythmView software (Abbott EP), a common visualization tool in the clinical practice. NIOM recordings were preprocessed in a custom MATLAB program to apply band-pass filtering 0 to 64 Hz, pixel binning 3 by 3, and normalization in the range 3-98% of the total intensity. Maps of heart electrical activity had the size of 100×100 pixels and were cropped to show the same field of view as the MEM maps. This resulted in 1200 pairs of images for training the generative model and 400 images for the evaluation.

**Experiments and results** We trained CycleGAN model on the pre-processed dataset of 5 explanted hearts for generating synthetic NIOM maps, testing it on 2 *ex-vivo* and 2 *in-vivo* cases. The image-to-image experiments were conducted in Python using the PyTorch framework and were run on Nvidia GeForce GTX 1080 Ti GPU 11GB VRAM. Structure Similarity (SSIM) between simultaneous MEM and synthetic NIOM images, and a human expert test (for measuring the medical relevance of the synthetic movies) were used to evaluate the results. Two clinical experts were asked to label the locations of the AF

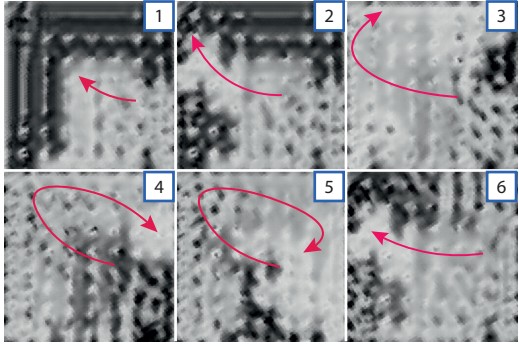

Figure 2: AF driver by synthetic images

| Heart | SSIM | $R_{Dice}$ |
|---|---|---|
| *ex-vivo* 1 | $0.56 \pm 0.06$ | $1.37 \pm 0.06$ |
| *ex-vivo* 2 | $0.60 \pm 0.04$ | $2.66 \pm 1.62$ |
| *in-vivo* 1 | $0.58 \pm 0.05$ | $1.35 \pm 0.22$ |
| *in-vivo* 2 | $0.58 \pm 0.04$ | $1.38 \pm 0.25$ |

Table 1: Metrics for different hearts

reentrant drivers in movies from the real MEM and the synthetic NIOM images, which were compared with those of the ground-truth region by the Dice metric. We naturally proposed a coefficient $R_{Dice} = Dice_1/Dice_2$ to quantify the human expert test, where $Dice_1$ is the score between the ground-truth and the synthetic NIOM masks and $Dice_2$ is that between the ground-truth and the real MEM masks. We showed that the generated images are similar to the real MEM ones, with the mean SSIM of 0.58. The metrics are presented in Table 1. The calculated ratios of Dice scores are higher than 1 in all cases, meaning that it is easier for the experts to correctly identify the AF driver region (*e.g.*, see Fig. 2) using the synthetic high-resolution NIOM movies rather than using the real MEM movies.

**Conclusions** In this study, we addressed the problem of poor visualization of the AF source patterns in clinical cardiac imaging. We suggested a new paradigm of *ex-vivo–*to*–in-vivo* learning, where an image-to-image model is trained on *explanted* organs and is applied to the real clinical cases. We demonstrate that the generated synthetic images resemble the real ones both visually and quantitatively. Furthermore, the limited annotation results by the clinical experts show that the AF rotational pattern is better visualized in synthetic movies rather than in the original real clinical videos. Large-cohort validation and extension to other explanted organs should further support the proposed learning paradigm.

## Acknowledgments

We thank the Lifeline of Ohio for providing the explanted hearts. The study was funded by NIH grants HL135109, HL115580 and the Bob and Corrine Frick Center for Heart Failure and Arrhythmia (Prof. Fedorov) and RFBR grant 19-29-01240 "mk" (Prof. Dylov).

## References

Jun-Yan Zhu, Taesung Park, Phillip Isola, and Alexei A. Efros. Unpaired image-to-image translation using cycle-consistent adversarial networks. *CoRR*, abs/1703.10593, 2017.

Alexander Zolotarev, Brian J. Hansen, Ekaterina Ivanova, Katelynn Helfrich, et al. Optical mapping-validated machine learning improves atrial fibrillation driver detection by multi-electrode mapping. *Circ Arrhythm Electrophysiol*, 13(10):008249, 2020.

