# OpenReview forum: "Ex-vivo - to - In-vivo Learning in Cardiology"
_MIDL.io/2021/Conference/Short — MIDL 2021 Poster_

### Official Review · Reviewer_i5DD · 2021-04-20

**Confidence:** 4
**Final Rating:** 3

**Summary:**

This paper presents a very interesting application of unsupervised image-to-image translation using CycleGAN. Specifically, they propose to use NIOM images acquired with high detail ex vivo in order to enhance lower quality MEM images acquired in vivo. These images are used to identify regions driving atrial fibrillation (AF). In validation, they use direct (SSIM) and indirect measures (Dice scores of these driving regions, drawn on the different modalities). The authors show that it's easier for an expert to label the regions using the synthetic NIOM images, rather than the real MEM movies.

**Strengths:**

This is a very novel, clinically relevant, and very interesting application of CycleGAN.

The evaluation does not stop at image similarity metrics, but goes all the way to the real target ("can I label things better in these enhanced images").


**Weaknesses:**

The sample size is small, but being explanted hearts, it's more than understandable.

The Dice scores and SSIM scores are modest, but this is a very difficult problem (simultaneous and unsupervised super-resolution and synthesis).

Lack of comparison with other methods, such as CycleGAN regularized by, e.g., segmentation.

**Deanonymize Review:**

no

**Detailed Comments:**

If the ultimate goal is identifying the AF driving regions, why not trying to segment them directly with eg a Unet? You could train on MEM images with ground truth derived from NIOM.

Since there is a know resolution gap between the two modalities, why not incorporating in into the NIOM->ME branch of the CycleGAN (i.e., with an explicit blurring/subsampling layer)?

**Justification Of The Rating:**

I think the MIDL community would be interested in this difficult, novel application of unsupervised synthesis (and super-resolution!). I think the small sample size is justified given the nature of the data. The lack of comparison is definitely a minus but, altogether, I think that the audience would enjoy this presentation.

**Paper Type:**

validation/application paper

**Special Issue:**

no

---

### Official Review · Reviewer_dJSR · 2021-04-29

**Confidence:** 4
**Final Rating:** 4

**Summary:**

The authors present a GAN-bases synthesis method for increasing the resolution of multi-electrode mapping (MEM) for atrial fibrillation assessment. Thus, they propose image-to-image translation from a low-resolution MEM image to a high-resolution near-infrared optical mapping (NIOM) image while sparing the visible structures from the MEM.
The evaluation is done on a set of 7 ex-vivo human hearts where both kinds of images and hence a ground truth are available. Furthermore, the method was evaluated on 2 in-vivo cases without a ground truth, but with expert labels of the AF drivers.

**Strengths:**

The problem tackled in this study is highly relevant for clinical application. Even though the general idea of translating a course image to a different modality with a higher resolution using GANs is not new, its application in the context of AF mapping seems to be novel. Regarding its complexity, the evaluation is good and reliable. The results are promising and the metric $R_{Dice}$ is a neat idea to show the improvement over the classical method.

**Weaknesses:**

The typical concern with synthesizing high dimensional data from a low-dimensional surrogate in a clinical context is the consistency of the data, and this study is no excuse. However, by performing the expert labeling experiment, the authors could show that their approach improves the surgeons accuracy. Additionally, they explicitly state that their aim is to design a better visualization for the physicians, which matches the results.

The dataset, specifically the in-vivo data, is too small to be reliable. However, due to the high effort in collecting such data and the fact that this is a short paper, I would count this as an issue for "Future Work".

**Deanonymize Review:**

yes

**Detailed Comments:**

The paper lacks some important details on the data processing (pixel binning, normalization, etc.). It would be nice to have parameter values given in the paper.

**Justification Of The Rating:**

This study presents a novel method of high clinical relevance. Within the scope of a short paper and regarding the complexity of data collection, the evaluation is sound and the results are very promising.
I am sure this work will complement the MIDL 2021 and will raise interesting discussions on image synthesis as well as the evaluation of ex-vivo to in-vivo transfer approaches.

**Paper Type:**

validation/application paper

**Special Issue:**

yes

---

### Meta-Review · Program_Chairs · 2021-05-09

**Recommendation:** Accept (Poster)
**Confidence:** 4

**Metareview:**

Reviewers are unanimous in their recommendation to accept this paper.

---

### Decision · Program_Chairs · 2021-05-11

Accept (Poster)